# Revisiting the Interaction of γδ T-Cells and B-Cells

**DOI:** 10.3390/cells9030743

**Published:** 2020-03-18

**Authors:** Francesca Rampoldi, Leon Ullrich, Immo Prinz

**Affiliations:** Institute of Immunology, Hannover Medical School, 30625 Hannover, Germany; ullrich.leon@mh-hannover.de (L.U.); prinz.immo@mh-hannover.de (I.P.)

**Keywords:** γδ T-cells, B-cells, γδ T–B-cell interaction, IL-4, autoantibodies

## Abstract

Right after the discovery of γδ T-cells in 1984, people started asking how γδ T-cells interact with other immune cells such as B-cells. Early reports showed that γδ T-cells are able to help B-cells to produce antibodies and to sustain the production of germinal centers. Interestingly, the presence of γδ T-cells seems to promote the generation of antibodies against “self” and less against challenging pathogens. More recently, these hypotheses were supported using γδ T-cell-deficient mouse strains, in different mouse models of systemic lupus erythematous, and after induction of epithelial cell damage. Together, these studies suggest that the link between γδ T-cells and the production of autoantibodies may be more relevant for the development of autoimmune diseases than generally acknowledged and thus targeting γδ T-cells could represent a new therapeutic strategy. In this review, we focus on what is known about the communication between γδ T-cells and B-cells, and we discuss the importance of this interaction in the context of autoimmunity.

## 1. Introduction

The adaptive immune system of vertebrates relies on three different lymphocyte lineages. B-, αβ T-, and γδ T-cells generate three different forms of clonal antigen receptors, namely the B-cell receptor (BCR), the αβ T-cell receptor (αβ TCR), and the γδ TCR, respectively.

The γδ TCR was first cloned in 1984 and led to the discovery of γδ T-cells as an independent T lymphocyte lineage [1,2,3]. They represent a small percentage (1–5%) of total T-cells in human blood and express a TCR heterodimer composed of a γ and a δ chain [4,5,6]. γδ T-cells are enriched in epithelial tissues such as skin, gut, and lung, but they are also found in secondary lymphoid organs like spleen and lymph nodes.

γδ T-cells represent a bridge between the innate and the adaptive immune system: on one side, they can respond with relatively high clonal frequency and rapidly produce a variety of cytokines at the beginning of an immune response without the requirement of prior activation and clonal expansion. On the other side, they can behave like the adaptive arm of the immune response because of their pleiotropic effector functions and their clonal expansion after being primed via their TCR [7,8].

αβ and γδ T-cells derive from the same common lymphoid progenitor in the thymus. The development of γδ T-cells starts with the productive rearrangement of their *Trg* and *Trd* loci. Failure to produce a functional γδ TCR drives T-cell progenitors with a functional TCR β-chain to enter the CD4^+^CD8^+^ double-positive (DP) stage where they rearrange their *Tra* loci and eventually display a functional αβ TCR. [9]. Not much is known about the interplay between αβ and γδ T-cells during their development. However, DP αβ T-cell progenitors can interact with early γδ T-cell progenitors and can condition the development of interferon-γ (IFN-γ)-producing γδ T-cells. This process is called *trans-*conditioning and depends on signals received through the lymphotoxin-β receptor, which serves pleiotropic functions including chemokine, cytokine or adhesion molecule expression, cell proliferation, and cell survival [10,11,12]. More recently, it has been shown that γδ T-cells can also influence the homeostasis of post-thymic CD4^+^ and CD8^+^ αβ T-cell populations, in particular of cells with a memory phenotype [13]. Rather than via direct interaction between the two kinds of T-cells, this effect seems to be due to the production of IL-4 by αβ T-cells, which is in turn strictly regulated by the γδ T-cells [13].

In mice, Vγ1^+^ and Vγ4^+^ T-cells are mainly found in the secondary lymphoid organs. They are both very heterogeneous populations and can exert different functional effects in various mouse models of disease [14]. Vγ1^+^ T-cells are able to produce IFN-γ, IL-4, and IL-13, and a small subpopulation behave like NKT cells [15,16]. Vγ4^+^ T-cells can produce IL-17, TNF, IFN-γ, and IL-4, according to the studied model [16]. In particular, in a mouse model of collagen-induced arthritis, which shares many hallmarks with human rheumatoid arthritis, Vγ4^+^ cells are activated and produce IL-17 [17]. Moreover, they are able to promote viral myocarditis and virus-induced lung inflammation by releasing IFN-γ and TNF-α [18,19].

Human γδ T-cells are subdivided based on the V-gene usage of the TCR δ chain. The most abundant and studied subpopulations in the human blood are Vδ1^+^ and Vδ2^+^ cells, whereas cells expressing other TCR δ chains are less frequent [20]. Vδ1^+^ T-cells are mainly localized at mucosal surfaces but can also be found in the blood, while Vδ2^+^ T-cells represent the main γδ T-cell subpopulation in the peripheral blood [21]. In most of the Vδ2^+^ T-cells, the Vδ2-chains pair with Vγ9-chains and this population can increase up to >50% of blood lymphocytes after certain bacterial or parasitic infections [22,23].

Current research on γδ T-cells focuses on discovering their TCR ligands in order to understand their functions and interactions within their tissue of residence. Still, an intriguing question is how γδ T-cells interact with other immune cells. In this review, we will focus primarily on the interaction between B and γδ T-cells and the contribution of γδ T-cells to humoral immunity and autoimmunity.

## 2. How Do γδ T-Cells Influence B-Cells?

It is well known that γδ T-cells have a strong impact on humoral immunity [24,25,26,27]. B-cells differentiate into antibody-(Ab)-producing plasma cells through a series of developmental steps that start in the bone marrow. Here, in the bone marrow, γδ T-cells do not seem to be involved in B-cell development [28].

Once B-cells express a rearranged BCR on the surface, they are negatively selected for self-antigen in the bone marrow during the first checkpoint of B-cell tolerance. Next, they leave the bone marrow and migrate via the bloodstream to the spleen. There, immature B-cells mainly differentiate through transitional stages into follicular and marginal zone (MZ) B-cells. By using γδ T-cell-deficient mouse strains, it has emerged that peripheral B-cells are influenced by the absence of γδ T-cells. Specifically, transitional, MZ B-, follicular B-, and the so called aged-associated B-cells (ABC)s were affected, indicating the capability of γδ T-cells to regulate precise stages of B-cell development [28]. All these B-cell subpopulations were decreased in the absence of Vγ4^+^ T-cells due to the altered γδ T-cell compartment and its inability to produce IL-4. Interestingly, direct contact of MZ B-cells with γδ T-cells seems to be responsible for their decrease [28].

After encountering their specific antigen, activated B-cells interact with their cognate T follicular helper (Tfh) cells, a subset of αβ T-cells which provide help to B-cells, resulting in their differentiation into extrafollicular short-lived plasma cells or their migration into the follicles to form germinal centers (GC)s [28,29,30,31,32]. This interaction involves the binding of costimulatory molecules like CD40L, Inducible T-cell costimulator (ICOS), and CD28 expressed on the γδ T-cells to their B-cell counterpart CD40, ICOS-L, and CD86, and the production of IL-21 by Tfh cells for their maintenance [33,34,35]. Within the GC, B-cells undergo isotype switching and somatic hypermutation of their immunoglobulin (Ig) genes to increase BCR affinity. Affinity-matured GC B-cells are then selected to differentiate into long-lived memory B-cells or plasma cells [36].

So, how could γδ T-cells potentially influence post bone marrow B-cell maturation and extrafollicular/GC B-cell differentiation into plasma cells? A number of interactions and mechanisms are conceivable (Figure 1): γδ T-cells may (i) assume a role similar to Tfh [25,37], (ii) influence the development of Tfh via Wnt ligands [38], (iii) produce chemokines like CXC chemokine ligand 13 (CXCL13), which is important for the migration of B-cells into the spleen follicles [39], (iv) serve as professional antigen presenting cells (APC)s in the priming of antigen specific B-cells, (v) release cytokines such as IL-4, which is important for B-cell development [40], or (vi) imprint APCs and therefore influence the nature of Tfh–B-cell interaction [41].

A subset of human Vγ9^+^Vδ2^+^ T-cells isolated from peripheral blood expresses the CXC chemokine receptor type 5 (CXCR5) like Tfh cells, and, upon antigen stimulation, they are able to express the costimulatory molecules ICOS and CD40L, to produce cytokines such as IL-2, IL-4, and IL-10, and to help B-cells for the production of Abs [25]. This Vγ9^+^Vδ2^+^ T-cell subset expresses CXCL13, CXCR5, and ICOS and provides help to B-cells in the presence of the phosphoantigen (E)-4-hydroxy-3-methyl-but-2-enyl pyrophosphate (HMB-PP) and the cytokine IL-21 [26].

More recently, it has been shown that Vδ3^+^ T-cells can also upregulate CD40, CD86, and HLA-DR and promote the production and the release of IgM, but not IgG, IgA, or IgE by B-cells [42]. Together, these studies suggest that certain γδ T-cells, upon determinate conditions, can acquire a role similar to the Tfh [24].

Besides their direct role in helping B-cells, γδ T-cells can also regulate Tfh cells. Rezende and colleagues showed that *Tcrd^−/−^* mice immunized with ovalbumin presented far less Tfh cells compared to wild-type (WT) mice, suggesting a role for γδ T-cell in the development of Tfh. Indeed, they discovered, for the first time in mice, that a subpopulation of γδ T-cells expresses CXCR5, and, by releasing Wnt ligands, these cells are able to initiate the Tfh cell program in CD4^+^ cells. Interestingly, this γδ T-cell subpopulation can function as an APC to naïve T-cells [38].

In humans, phosphoantigen-activated Vγ9^+^Vδ2^+^ T-cells display the main characteristics of a professional APC, they efficiently process and display the antigens on MHCII molecules, and provided co-stimulatory signals for strong induction of naïve CD4^+^ T-cell proliferation and differentiation [43].

IL-4 is a typical signature cytokine of the type II inflammatory response triggered during parasitic infections and allergy. IL-4 can be produced by CD4^+^ T, γδ T, NKT, B-cells, basophils, eosinophils, mast cells, and also by type-2 innate lymphoid cells. In mice, IL-4 induces the differentiation of naïve CD4^+^ T-cells into Th2 cells, drives the Ig class switch to IgG1 and IgE in B-cells, and induces alternative macrophage activation [44]. IL-4 can also induce Ig class switching toward the expression of IgG4 and IgE in humans [45,46].

Early experiments conducted in mice that congenitally lack αβ T-cells showed that their B-cells could still expand and secrete Abs of the subclasses IgG1 and IgE, suggesting for the first time a role for IL-4 producing γδ T-cells in helping B-cells [40]. Similarly, IgG1 and IgE were highly increased in the serum of mice deficient of the Vγ4^+^ and Vγ6^+^ T-cell populations, which also presented increased levels of IL-4 in the serum [47]. These *Vγ4*^−/−^/*6*^−/−^ mice were then crossed with IL-4 KO mice to remove any possible effect of IL-4, whose production is enhanced by Vγ1^+^ cells that developed in the absence of Vγ4^+^. The lack of IL-4 in the double mutant mice abrogated the phenotype of *Vγ4*^−/−^/*6*^−/−^ mice, indicating an important role of γδ T-cells in the regulation of IL-4 and Ig production [47].

More recently, Crawford and colleagues studied a model of epithelial damage in the skin induced by using the DNA-damaging xenobiotic and carcinogen DMBA (7,12-dimethylbenz[a]anthracene) [48]. In this model, γδ TCR^+^ intraepithelial lymphocytes (IEL)s could trigger the production of IgE, which confer protection in the exposed tissue. Interestingly, this endogenous IgE response was autoreactive and supported by IL-4-producing CD4^+^ αβ T-cells. In addition, sequencing of IgE antibodies in WT and *Tcrd*^−/−^ mice revealed that *Tcrd*^−/−^ mice had a defective clonal expansion of IgE^+^ B-cells and a less diverse B-cell repertoire compared to WT mice. Intriguingly, when using another chemical agent, TPA (12-O-tetradecanoylphorbol-13-acetate), which does not induce DNA damage but triggers skin inflammation, the induced IgE antibodies were not autoreactive. In summary, γδ TCR^+^ IELs could shape a distinct IgE repertoire that was dependent on the presence of DNA damage in epithelial cells, indicating a unique role of γδ TCR^+^ IELs in initiating and regulating IgE production and repertoire [48].

## 3. How Do B-cells Influence γδ T-Cells?

So far, we have focused on the influence of γδ T-cells on B-cells, but what about vice versa? Already in 1989, Sperling and Wortis found that a syngenic B-cell lymphoma induced proliferation of murine γδ T-cells after co-culture [49]. Similarly, Daudi cells, a human Burkitt′s lymphoma cell line, have been extensively studied in the presence of human Vγ9^+^Vδ2^+^ T-cells as they are able to activate specific γδ TCRs [50,51,52]. Interestingly, human Vδ1^+^ T-cells also seem to be stimulated by activated B-cells isolated from peripheral blood [53,54]. Moreover, the expression of B7 and CD39 molecules on the surface of activated B-cells appears to be important for the activation of human Vδ1^+^ T-cells [53]. Therefore, the putative γδ T-cell-stimulatory ligand on B-cells is most likely of cellular rather than viral origin, and its expression is upregulated upon activation of B-cells. In conclusion, the influence of B-cells on γδ T-cells seems to be more TCR ligand-dependent, which is not the main topic of this review but has been thoroughly reviewed elsewhere [55].

Finally, B-cells may interact with γδ T-cells via butyrophilins (BTN)s. BTNs are members of the B7 family of costimulatory receptors, which include B7.1 (CD80) and B7.2 (CD86), and have also been recognized as potentially important immune modulators [56]. They are pivotal in the maintenance of immune homeostasis in tissue epithelium, by controlling the selection/activation of γδ T-cells, such as Skint1 for Vγ5^+^Vδ1^+^ T-cells [57]. More recently, BTN and BTN-like molecules have emerged not only as selecting agents for developing γδ T-cells but also direct ligands of human γδ TCRs (Vγ9^+^Vδ2^+^ T-cells and Vγ4^+^ T-cells) [58,59,60,61,62]. BTN2A2, a BTN family member that has not yet been implied in γδ TCR recognition, is expressed on the surface of B-cells in the spleen, suggesting a possible role in γδ T-cell recognition and homeostasis [63].

In order to better study the effect of B-cells on γδ T-cells, it is important to find the right experimental approach. Previously, a common way to investigate the function of γδ T-cells was the depletion of γδ T-cells via the use of monoclonal antibodies anti-γδ TCR which block and internalize the γδ TCR [64,65]. The problem with mAbs is that they do not lead to a depletion of γδ T-cells in vivo, but rather to the internalization of the γδ TCR and the generation of “invisible” γδ T-cells [66]. The use of *Tcrd*^−/−^ mice constitutively depleted in the δ chain [67] can be useful in loss-of-function experiments; surprisingly, these mice display a relatively mild phenotype [68,69]. This can be due to the fact that αβ T-cells can fill up the niche of γδ T-cells and can partly fulfill their function [70]. A new way to analyze γδ T-cells is the use of *Tcrd*-GDL mice where the γδ T-cells can be conditionally depleted through the injection of diphtheria toxin (DTx) as the mice express a human DTx receptor. Therefore, this model will allow the depletion of γδ T-cells without the compensatory effects of αβ T-cells and compromising other αβ T-cells [71].

## 4. γδ T-Cells and Germinal Centers

The GC reaction plays an important role in humoral immunity, as B-cells with the highest affinity to GC-contained antigens are selected to become plasma cells that produce high-affinity antibodies or memory B-cells [72,73,74]. GCs are formed in secondary lymphoid organs at different time points. In the spleen and lymph nodes, GCs are formed during an infection or vaccination [36]. In contrast, in Peyer’s patches in the gut, GCs are formed constitutively through the stimulation of the gut microbiota [36,75]. For an effective GC reaction, the interaction of various lymphocyte types such as B-cells, follicular dendritic cells (FDC)s, αβ T-cells (especially Tfh), and most likely also γδ T-cells, is required.

Early studies have shown that γδ T-cells can be found inside GCs [76,77]. Furthermore, γδ T-cells could be localized in the GCs of the gut mucosa, lymph nodes, and spleen of control patients as well as in patients with Yersinia infection and Crohn´s disease [78]. However, the role of γδ T-cells in the GC is still not clear.

Hayday and collaborators showed in the 1990s that GCs can be formed in *Tcra*^−/−^ and *Tcrb*^−/−^ mice (i.e., αβ T-cell-deficient but γδ T-cell-competent mice) upon infection [79,80]. Subsequently, it turned out that depletion of specific subsets of γδ T-cells had a bigger impact than complete depletion of all γδ T-cells [81]. Mice that carried only Vγ4^+^ and Vγ6^+^ γδ T-cells showed a decrease in total Ig level in sera. In contrast, mice carrying only Vγ1^+^ γδ T-cells showed an increase of total Ig level and displayed spontaneous GCs in the spleen [47]. As mentioned above, γδ T-cells can influence the GC reaction also through the regulation of Tfh cells. Accordingly, *Tcrd*^−/−^ mice immunized with CFA and OVA showed reduced frequencies of GC B-cells due to the missing help of the Tfh cells [38].

Further evidence that γδ T-cells play a role in providing help to B-cells was found in experiments involving human blood Vγ9^+^Vδ2^+^ T-cells. These γδ T-cells stimulated with phosphoantigens could provide B-cell help through cell–cell contact. Within 36 h after stimulation, CD40L, OX40, and ICOS were highly upregulated, and, in co-culture experiments, they acted as strong B-cell help for production of IgM-, IgG-, and IgA-like Tfh cells [78]. In the same manner, Vγ9^+^Vδ2^+^ T-cells activated with the phosphoantigen HMB-PP and in presence of IL-21, which is mainly produced by Tfh in the GC, showed production of CXCL13. This chemokine is normally produced by FDCs in the light zone of the GC in order to guide B-cells into the light zone [39,82,83]. αβ T-cells have the ability to recirculate continuously between secondary lymphoid organs. In contrast, γδ T-cells start with the upregulation of homing receptors for secondary lymphoid organs just after their activation. This phenomenon may indicate that γδ T-cells play a role in early phases of infection and support the GC reaction, before αβ T-cells could provide further help [78].

Together, these findings suggest an involvement of γδ T-cells in B-cell class-switching (extra-follicular or within the GC) and in supporting GC reactions at different stages [84]. Additionally, γδ T-cells could also play a role in the maintenance or survival of plasma cells [28]. However, in most of the cases, the exact mechanisms have not been clearly identified, showing that further research has to be done.

## 5. γδ T-cells and Autoimmune Disease

After reviewing the possible ways of γδ-B-cell interaction, the next question is why γδ T–B-cell interaction would be important. Clearly, there is a positive association between γδ T-cells and autoimmunity [85,86]. However, the role of γδ T-cells for the production of autoantibodies and breaking tolerance still remains unclear.

What it is known so far from *Tcra*^−/−^ mice is that their serum antibodies are more autoreactive than those of WT control mice. In particular, they are positive for anti-nuclear antibodies (ANA)s, anti-double stranded (ds)-DNA and small nuclear ribonucleoproteins which are typical autoantibodies that are characteristic of systemic lupus erythematosus (SLE) [40,87]. Similar autoantibodies have been detected in Vγ4/6-deficient mice, which present an increase of anti-single stranded and anti-ds DNA, and anti-nuclei autoantibodies. The presence of these autoantibodies is probably explained by the ability of the remaining Vγ1^+^ T-cells and, mainly, αβ T-cells (regulated by the γδ T-cells) for producing large amounts of IL-4 that activate B-cell maturation and can overrun tolerance mechanisms [13,15,47]. IL-4 production may be implicated in the breakdown of B-cell tolerance by promoting the survival of BCR-mediated apoptosis of splenic B-cells and transitional B-cells during negative selection [88,89,90]. In addition, *Tcrb^−/−^* mice were able to generate self-reactive antibodies after parasitic infection, in particular towards DNA instead of antibodies specific for the pathogen, thereby supporting the idea that γδ T-cells are more important for autoantibody production rather than mounting a pathogen-specific immune reaction [79].

Recently, an autoantibody microarray was performed on serum from WT and *Tcrd*^−/−^ mice at steady state and after induction of a murine model of SLE. *Tcrd*^−/−^ mice showed decreased autoantibody production at steady state and upon induction of SLE [38]. Possible explanations of the recurrence of all these autoantibodies can be due to the fact that γδ T-cells may help polyclonally activated B-cells [87] or that γδ T-cells may present autoantigens to B-cells [43]. At this moment, it is hard to speculate about the mechanisms involved, but future studies will probably shed light on this mystery.

Thus, γδ T-cells seem to play an important role in the regulation of human autoimmune diseases such as inflammatory bowel disease and experimental autoimmune encephalomyelitis [85]. Moreover, they have a strong clinical association with many autoimmune diseases like rheumatoid arthritis and SLE. Several studies reported that γδ T-cells were present in significantly higher number in SLE patients compared to healthy controls [27,91]. Therefore, targeting the interaction of γδ T- and B-cells may be an attractive therapeutic strategy for the prevention of autoimmunity.

## 6. Conclusions

γδ T-cells seem to have the potential to regulate B-cell maturation during their development in the periphery (spleen) and during an immune response (in GC). Whether such an influence of γδ T-cells is mediated via soluble mediators important for B-cells (such as IL-4), or rather by the presentation of autoantigens, remains to be determined [24]. At this moment, our knowledge of the influence of γδ T-cells on B-cells remains limited. It is also clear that there is a correlation between γδ T-cells and autoimmune disease. Discovering the mechanism/s behind this correlation would help to understand the causes of these diseases and could potentially help to design novel therapeutic strategies

## Figures and Tables

**Figure 1 cells-09-00743-f001:**
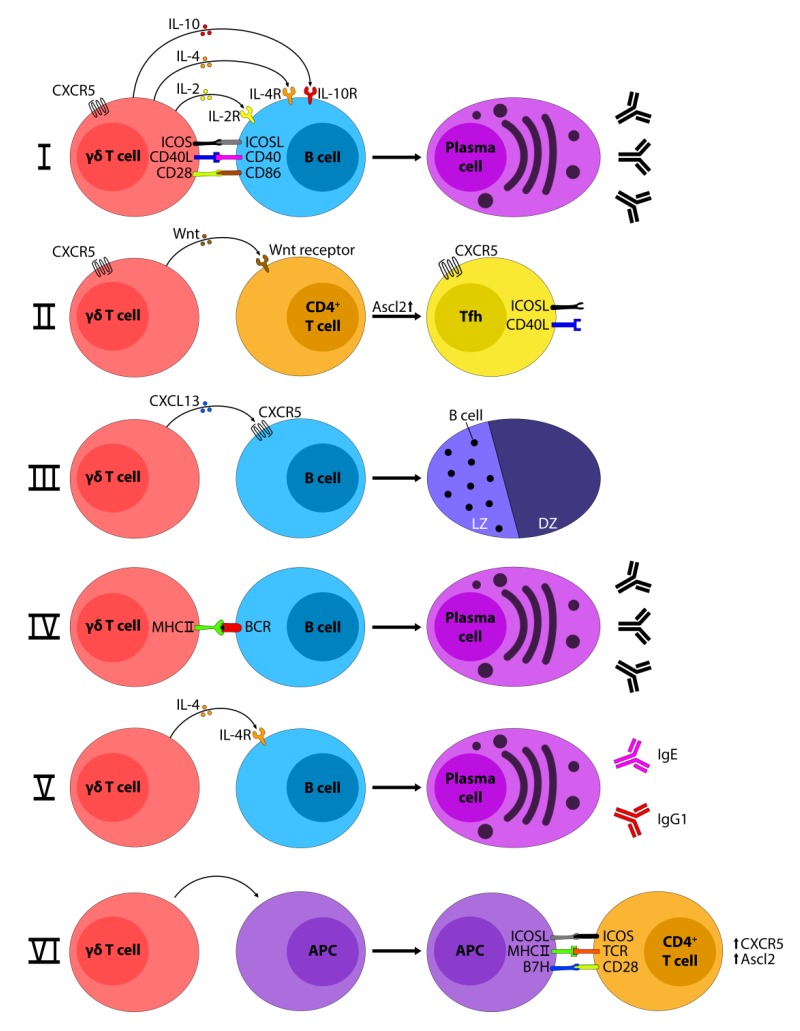
Possible γδ T–B-cell interactions. γδ T-cells can interact with B-cells via costimulatory molecules like CD40/CD40L, Inducible T-cell costimulatory (ICOS)/ICOSL, and CD28/CD86, and release cytokines like IL-2, IL-4, and IL-10 that favor the development of B-cells into plasma cells (**i**,**v**). γδ T-cells can as well promote the development of CD4^+^ T-cells into T follicular helper (Tfh) by the release of Wnt molecules (**ii**). Besides the development, γδ T-cells can also influence the localization of B-cell inside the germinal center, positioning them into the light zone thanks to the production of CXC motif chemokine 13 (CXCL13) (**iii**). Finally, γδ T-cells may act as antigen presenting cells (APC)s and present the antigen to B-cells (**iv**) or they may imprint other APCs like dendritic cells, which, after binding with CD4^+^ αβ T-cells, initiate the Tfh development (**vi**). Ascl2, Achaete-scute complex homolog 2; BCR, B-cell receptor; CD40L, CD40 ligand; CXCR5, CXC-chemokine receptor 5; DZ, dark zone; ICOSL, ICOS ligand; Ig, immunoglobulin; IL-4R, IL-4; receptor; LZ, light zone.

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
