# Peer review of "Revisiting the Interaction of γδ T-Cells and B-Cells"

_cells, 2020, doi:10.3390/cells9030743_

Round 1
Reviewer 1 Report
The review by Rampoldi et al. represents a very nice introduction into the timely topic of gd T - B cell-interactions. As stated by the authors, this topic has been "on the radar" of researchers almost sine inception of the field. Many questions remain, and the respective roles of ab T cells and gd T cells as modulators of B cell development and function are not yet fully defined. Likewise, the influence of B cells on these T cell populations also deserves further study.
The short review covers many of the key issues. A few details might be worth mentioning still:The authors state that "not much is known about the interplay between ab and gd T cells during their development" (lines 48/49), but besides the work of the Hayday group, a recent study by Swati Phalke et al. in PLOS1 seems relevant to this question.
Lines 59/60: "But how gd T cells influence B cells?" I suggest to add a "do" to make the question: But how do gd T cells influence B cells?
Lines 68/69: Actually, the paper referenced as well as ref#19 describe large gd T cell-dependent changes not only in transitional and marginal zone B cells but also in follicular B cells and "new" B cells, also known as ABCs.
Lines 219-21: As addressed in the Phalke paper, the major source of IL-4 appear to be ab T cells after all, with gd T cells regulating the output of ab T cells. Although NKT-like Vg1+ cells are capable of producing large quantities of IL-4, they are few in number when compared to IL-4 producing ab T cells and seem to contribute only a minor portion of the IL-4 in circulation.
Reviewer 2 Report
This is a fascinating topic for a review and timely to revisit what is known in this area. The review is generally well conceived and touches on most known studies relating to gd T cell/B cell interactions. I have no major concerns regarding the publication of this review and welcome renewed interest in the area.
The English language may need a little tidying up and the authors should check through for correct use of present/past tense and singular/plural. I may mention a few changes below but in general have not corrected/suggested change of language. A few suggestions/comments here:
- section starting line 45: perhaps better earlier in the intro?
- line 54:...discovering their TCR ligands...
- line 59-60: incomplete sentence:'But how gd T cells influence B cells?'
- line 71:....a subset of ab T cells which...
- line 78 and generally: the authors should consider that gd T cells may not only influence B cells in GCs. Class-switching can occur extra-follicularly, which gd T cells may support, and gd T cells may also be important for example in the maintenance/survival of plasma cells
- Figure 1: gd T cells could also influence B cells indirectly for example by imprinting DCs/APC to ultimately determine the nature of the abT cell - B cell interaction
- line 125: 'Removal of IL-4....' This sentence needs clarity.
- line 139:'....and regulating the IgE production and repertoire'
- section starting line 141: important to state which gd T cells the authors are talking about in the different examples.
- line 152: BTNs are members of the B7 family
- line 155:...by controlling the selection/activation...
- line 183-185: the experiment described shows that gd T cells can provide B cell help - not that they support GCs. CD40L etc is also needed for extra follicular B cell activation/class switching...
- line 198: section starting here seems a little out of place/does not serve an obvious purpose here? I suggest to move or simply remove.
- line 211: 'Clearly there is an association...' Clarify what you mean here - positive or negative association or?
- line 220: hoe does IL-4 production 'overrun' tolerance mechanisms?
- line 224: autoantibodies are a specific immune reaction - maybe rather: a 'pathogen-specific immune reaction'?
- line 224: take away 'however'
Reviewer 3 Report
This is an interesting and timely review, and is quite well written. In a few places, the authors could expand on the context of the statements they are making. For example:
Line 39: which “studied model”? – it would be better to briefly describe the model than to make the reader look it up
Line 68: how are transitional and MZ B cells affected? Please just explain in a bit more detail.
Other minor points:
Line 41: There are other human γδ Τ cell subsets beyond Vδ1 and Vδ2, though they are the best studied. The authors themselves refer to Vδ3 cells elsewhere in this manuscript (line 101). In Adrian Hayday’s 2000 review in Annual Review of Immunology, he states that there are 8-10 Vδ genes and lists references: See Arden B et al Immunogenetics 1995 and Clark SP et al Immunogenetics 1995. Please just change the wording accordingly.
Please change the titles for sections 2 and 3 to:
- How do γδ T cells influence B cells?
- How do B cells influence γδ T cells?
Line 62: by “their development” do the authors mean B cell development? Be specific.
Lines 73-75: It would be helpful to indicate which receptors are on which cells, and the authors should reference Figure 1, although CD28/CD86 is missing from the figure. Perhaps consider adding this in?
Figure 1: Arrowheads are missing in the top panel (I)
Line 120: Please correct to IgG4; also, the authors have cited a review that cites the original research. Please cite the original research instead: Gascan H et al J Exp Med 1991.
Line 147: Since γδ Τ cells can be activated also in the absence of B cells, “crucial” is perhaps not the best word here; however, if the authors provide the context of the model in which this interaction appeared to be crucial, then it is fine to leave it in. Otherwise, it comes across as a general statement that is not quire correct.
English corrections/typos:
Line 42: add “cell” between “T” and “subpopulation”
Line 43: replace “rise” with “increase to” or “comprise”
Line 50: remove “s” from “cells”
Line 59: add “do” between “how” and “γδ T cells” OR, if you are changing the section title, omit this question altogether (redundant).
Line 60: add “plasma” between “producing” and “cells”
Lines 67 and 68: change “were” to “are”
Line 80: remove “the” between “via” and “Wnt”; add an “s” to “ligand”
Line 119: add “s” to “induce”
Line 127: remove “the” between “of” and “γδ”
Line 136: please insert “antibodies” between “IgE” and “were”
Line 147: remove “ed” from “appeared” to put the sentence in the present tense
Line 173: please change “1990’ies” to “1990s”
Line 185: remove “er” from “helper”
Line 188: remove “a”
Line 207: change “compensative” to “compensatory”
Lines 212/213: change “break of” to “breaking”
Line 216: remove ; and replace with “, which are”
Line 221: remove “the”
Line 230: please replace “enlighten” with “shed light on”
Line 239: change “of regulating” to “to regulate”
